# Peelable Alginate Films Reinforced by Carbon Nanofibers Decorated with Antimicrobial Nanoparticles for Immediate Biological Decontamination of Surfaces

**DOI:** 10.3390/nano13202775

**Published:** 2023-10-16

**Authors:** Gabriela Toader, Aurel Diacon, Edina Rusen, Ionel I. Mangalagiu, Mioara Alexandru, Florina Lucica Zorilă, Alexandra Mocanu, Adina Boldeiu, Ana Mihaela Gavrilă, Bogdan Trică, Daniela Pulpea, Mădălina Ioana Necolau, Marcel Istrate

**Affiliations:** 1Military Technical Academy “Ferdinand I”, 39-49 G. Cosbuc Blvd., 050141 Bucharest, Romania; gabriela.toader@mta.ro (G.T.); aurel_diacon@yahoo.com (A.D.); daniela.pulpea@mta.ro (D.P.); 2Faculty of Chemical Engineering and Biotechnologies, University Politehnica of Bucharest, 1-7 Gh. Polizu Street, 011061 Bucharest, Romania; alexandra.mocanu@upb.ro (A.M.); madalinanecolau@gmail.com (M.I.N.); 3Faculty of Chemistry, Alexandru Ioan Cuza University of Iasi, 11 Carol 1st Blvd., 700506 Iasi, Romania; 4Microbiology Laboratory, Horia Hulubei National Institute for R&D in Physics and Nuclear Engineering, 30 Reactorului St., 077125 Bucharest, Romania; mioara.alexandru@nipne.ro (M.A.); florina.zorila@nipne.ro (F.L.Z.); 5Department of Genetics, Faculty of Biology, University of Bucharest, 91-95 Splaiul Indepententei, 050095 Bucharest, Romania; 6National Institute for Research and Development in Microtechnologies—IMT Bucharest, 126A Erou Iancu Nicolae Street, 077190 Bucharest, Romania; adina.boldeiu@imt.ro; 7National Institute of Research and Development for Chemistry and Petrochemistry, 202 Splaiul Independentei, 060041 Bucharest, Romania; anamihaela.florea@gmail.com (A.M.G.); bogdan.trica@icechim.ro (B.T.); 8Advanced Polymer Materials Group, University Politehnica of Bucharest, 1-7 Polizu Street, 011061 Bucharest, Romania; 9S.C. Stimpex S.A., 46-48 Nicolae Teclu Street, 032368 Bucharest, Romania; marcel@yahoo.com

**Keywords:** carbon nanofibers, nanoparticles, nanosilver, antimicrobial activity, decontamination, alginate, nanocomposites

## Abstract

This study presents the synthesis and characterization of alginate-based nanocomposite peelable films, reinforced by carbon nanofibers (CNFs) decorated with nanoparticles that possess remarkable antimicrobial properties. These materials are suitable for immediate decontamination applications, being designed as fluid formulations that can be applied on contaminated surfaces, and subsequently, they can rapidly form a peelable film via divalent ion crosslinking and can be easily peeled and disposed of. Silver, copper, and zinc oxide nanoparticles (NPs) were synthesized using superficial oxidized carbon nanofibers (CNF-ox) as support. To obtain the decontaminating formulations, sodium alginate (ALG) was further incorporated into the colloidal solutions containing the antimicrobial nanoparticles. The properties of the initial CNF-ox-NP-ALG solutions and the resulting peelable nanocomposite hydrogels (obtained by crosslinking with zinc acetate) were assessed by rheological measurements, and mechanical investigations, respectively. The evaluation of Minimal Inhibitory Concentration (MIC) and Minimal Bactericidal Concentration (MBC) for the synthesized nanoparticles (silver, copper, and zinc oxide) was performed. The best values for MIC and MBC were obtained for CNF-ox decorated with AgNPs for both types of bacterial strains: Gram-negative (MIC and MBC values (mg/L): *E. coli*—3 and 108; *P. aeruginosa*—3 and 54) and Gram-positive (MIC and MBC values (mg/L): *S. aureus*—13 and 27). The film-forming decontaminating formulations were also subjected to a microbiology assay consisting of the time-kill test, MIC and MBC estimations, and evaluation of the efficacity of peelable coatings in removing the biological agents from the contaminated surfaces. The best decontamination efficiencies against *Staphylococcus aureus*, *Escherichia coli*, and *Pseudomonas aeruginosa* varied between 97.40% and 99.95% when employing silver-decorated CNF-ox in the decontaminating formulations. These results reveal an enhanced antimicrobial activity brought about by the synergistic effect of silver and CNF-ox, coupled with an efficient incorporation of the contaminants inside the peelable films.

## 1. Introduction

The recent worldwide medical crisis caused by the SARS-CoV-2 pandemic and the existing concerns about chemical warfare agents’ (CWA) potential utilization, underline the need for effective protective measures and decontamination procedures for managing biohazard or chemical threats [1]. Chemical and biological weapons, in contrast to nuclear or traditional weapons, solely target the human body and lead to temporary incapacitation or death. When a CBRN incident occurs, decontamination represents an enormous challenge that requires efficient resource management in the shortest time possible, with minimal waste generated [2].

This study mainly focuses on the remediation of biological contamination because the global problem of infectious and lethal bacteria-caused diseases is currently a serious scientific and medical challenge. Even if they are not classified as biological weapons (like *Bacillus anthracis*), five pathogens: *Staphylococcus aureus*, *Escherichia coli*, *Streptococcus pneumoniae*, *Klebsiella pneumoniae*, and *Pseudomonas aeruginosa* were responsible for 54.9% of deaths associated with bacterial pathogens in 2019 [3]. Reducing the burden of bacterial infection-related death is an essential global public health concern. To avoid the spread or re-aerosolization of biological agents, decontamination is essential. An extended study on cleaning efficacy has shown that using effective cleaning techniques can minimize airborne contaminants, occupant exposures, and health hazards [4]. While decontamination timeframes in the order of several hours may be reasonable for the civilian sector (microbiology laboratories, hospitals, schools, etc.), the military requires immediate intervention (30 min or less) [5]. Nevertheless, since biological contaminants pose supplementary threats due to their ability to multiply [6,7], finding fast and efficient decontamination methods is mandatory for any field of activity.

Decontamination might be as easy as washing with detergent, then rinsing with sterile water, then drying off [8]. Yet, fluids used for sanitization could also contain contaminants or work ineffectively [8]. Even the alcohol should be diluted with sterile water and put into sterile spray bottles, to be free of viable micro-organisms, including fungal and bacterial spores [8]. Hence, besides avoiding the use of potentially contaminated volatile solvents, using peelable coatings could bring multiple advantages, such as fewer steps required (applying the film-forming polymeric solution, followed by peeling after complete curing), inactivation and entrapment of the contaminants, a lower amount of post-decontamination waste, the possibility of avoiding solvents by utilizing aqueous polymeric solutions, etc.

Over time, strippable coatings have proved to be advantageous for the removal of contaminants from surfaces [9,10,11,12] representing a low-cost and effective method [9]. The coating formulations are designed to entrap contaminants via physical or chemical means [9]. Subsequently, when the coating is removed by peeling, the contaminants are transferred along with it. Peelable/strippable coatings are defined as temporary coatings that may be peeled off as continuous and sizeable films from the substrate after the service term [13]. Examples of commercially available peelable coatings used for decontamination are ALARA 1146, DeconGel 1108, TLC Free, Isotron Radblock, and InstaCote CC Wet/CC Strip [14]. It has been observed that the curing times vary, depending on the coating thickness, between 4 and 10 h for Stripcoat TLC Free™, 18 and 32 h for DeconGel^®^ 1108 or 4 and 24 h for CC Wet/CC Strip [15]. Long drying durations are one of the disadvantages of the casting method for producing films, which limits their application for decontamination [16]. UV-curable strippable films have recently gained popularity due to major advantages such as faster film formation time. However, UV light is required to cure these films, which can cause skin burns and can harm the cornea and retina [14]. Ionic crosslinking may represent a better option, since a complete curing can be achieved in a few minutes, depending on the thickness of the coating, and it involves lower health risks for the operators. The use of ionic crosslinked hydrogels can also be a more facile approach to the use of photoactive fabrics [17,18] designed for decontamination purposes, since the contact between the surface and active layer is assured by the water trapped at the end in the hydrogel.

Usually, these coatings are applied to the targeted surface via spraying, rolling, or brushing, and the coating is allowed to cure before being peeled away [9]. Depending on the polymeric matrix, the active ingredients, solvents, and compositions being employed, the coatings possess variable physical and chemical properties that make them suitable for a particular application [13]. Several types of peelable coatings have been developed for decontamination purposes [9,10,13,19,20,21,22,23,24,25,26,27]. However, there are fewer publications on biological decontamination of surfaces than on radioactive or CWA contaminants. The most common strategy to eliminate biological contaminants is to employ disinfectants (halogens, acids, alkalis, heavy metal salts, quaternary ammonium compounds, phenolic compounds, aldehydes, ketones, alcohols, and amines), but these compounds may be noxious and corrosive [28]. Thus, peelable coatings may offer a safer and more effective alternative.

Examples of polymers utilized for the formulation of strippable coatings include polyvinyl alcohol [24,29], polyethylene [24], polystyrene, poly (vinyl acetate) [9], polyacrylic acid [24], polyacrylates [30], polyaniline [31] polyurethanes [32], cellulose derivatives [33], N-halamine polymers [34] and other polysaccharides [35].

Reinforcing nanofillers are frequently used to obtain hydrogel matrices with greater mechanical resilience. Carbonaceous nanoscale materials, particularly carbon nanotubes (CNTs) and carbon nanofibers (CNFs) have attracted great interest as reinforcing agents [36,37]. For endowing these materials with antimicrobial properties, nanoparticles like silver, copper, nickel oxide, titanium dioxide, or zinc oxide, can be incorporated in the hydrogel formulations [38]. Faten Ismail Abou El Fadl et al. [39] described pectin/polyethylene oxide-based hydrogels containing 5 wt% nano-metal oxides (TiO_2_, CaO, MgO, and ZnO) synthesized through gamma irradiation, which demonstrated inhibitory effects against *P. mirabilis*, *S. aureus*, *P. aeruginosa*, and *C. albicans*. Qi Dong et al. [40] described hydrogels derived from a porcine dermal extracellular matrix incorporating silver nanoparticles (Ag-NPs) as wound dressings, demonstrating by in vitro and in vivo experiments the inhibition of bacterial growth. Yanjun Pan et al. [41] showed the advantages of utilizing poly (vinyl alcohol)/keratin films loaded with silver nanoparticles (Ag-NPs) as an alternative material for delayed chest closure applications, since these nanoparticles ensure the eradication/lower growth rate of *Staphylococcus aureus* and *Escherichia coli*.

Recent studies have been published on the improvement of the antibacterial properties obtained by combining carbon-based assemblies (single wall carbon nanotubes, multi walled carbon nanotubes, or graphene structures) with metallic or metal-oxide nanoparticles [42,43]. However, the potential of carbon nanofibers as a support platform for antimicrobial nanoparticles has been less explored.

This work aimed to develop hydrogel formulations designed for the fast neutralization, entrapment, and removal of biological agents from contaminated surfaces. To achieve this goal, the hydrogels must present antimicrobial activity and good mechanical properties to allow facile removal by pealing. Carbon nanofibers were selected as both reinforcing agents and as support for antimicrobial particles to be incorporated in the film-forming decontamination solutions. The presence of polar functional groups on the surface of CNF-ox serves as an anchoring point for the metal cations (by complexation) during the synthesis of the antimicrobial particles by reduction (silver and copper particles) and by precipitation (ZnO). The components displaying the highest antimicrobial activity were included and evaluated in the decontaminating formulation, to observe if they maintain their antimicrobial performances and afford the desired viscosity (wetting/spreading) and mechanical properties for the hydrogels after the crosslinking process.

This research brings into view a new approach for immediate decontamination applications by introducing innovative formulations that can quickly generate peelable hydrogels that can capture, inactivate, and incorporate biological contaminants without affecting the surface to which they are applied. The synergistic effect obtained by incorporating carbon nanofibers decorated with antimicrobial nanoparticles and the capacity of the composite polymeric matrix to entrap the biological contaminants ensure high decontamination efficiencies. Potential applications of the developed peelable alginate nanocomposite films may include the decontamination of biological agents, but the ease of tunability of these formulations may also allow the removal of other types of contaminants, i.e., by introducing mild oxidizers for the neutralization of chemical agents or chelating agents for the removal of heavy metals or radionuclides, etc.

## 2. Materials and Methods

### 2.1. Materials

*Materials used for the decontaminating formulations*: Polyvinyl alcohol (PVA, average Mw 85,000–124,000, 87–89% hydrolyzed, Sigma-Aldrich, St. Louis, MO, USA), carbon nanofibers (CNF, pyrolytically stripped, platelets (conical), >98% carbon basis, D × L 100 nm × 20–200 μm, Sigma-Aldrich, St. Louis, MO, USA), nitric acid (HNO_3_, 68%, Sigma-Aldrich, St. Louis, MO, USA), sulfuric acid (H_2_SO_4_, 95–98%, Sigma-Aldrich, St. Louis, MO, USA), sodium hydroxide (NaOH, Sigma-Aldrich, St. Louis, MO, USA), silver nitrate (AgNO_3_, Sigma-Aldrich, St. Louis, MO, USA), Trisodium citrate dihydrate (C_6_H_5_Na_3_O_7_·2H_2_O, Sigma-Aldrich, St. Louis, MO, USA), sodium borohydride (NaBH_4_, Sigma-Aldrich, St. Louis, MO, USA), copper acetate (Cu(CH_3_COO)_2_∙H_2_O, Sigma-Aldrich, St. Louis, MO, USA), ammonia (NH_3_, puriss., anhydrous, ≥99.95%,Sigma-Aldrich, St. Louis, MO, USA), ascorbic acid (C_6_H_8_O_6_, Sigma-Aldrich, St. Louis, MO, USA), zinc acetate (Zn(CH_3_COO)_2_∙2H_2_O, Sigma-Aldrich, St. Louis, MO, USA), sodium alginate (ALG, from brown algae, Carl Roth, Mw 300,000–350,000 g/mol, Karlsruhe, Germany), were used as received. Distilled water was employed as a solvent in all experiments.

*Materials used for the microbiology determinations included*: Muller Hinton broth (MHb) (Merck KGaA, Darmstadt, Germany), Mueller Hinton agar (Merck KGaA, Darmstadt, Germany), Phosphate-buffer saline (PBS, Merck KGaA, Darmstadt, Germany), Resazurin (resazurin sodium salt, powder, BioReagent, Sigma-Aldrich, St. Louis, MO, USA), LIVE/DEAD BacLight Bacterial Viability Kit (ThermoFisher Scientific Waltham, MS, USA), Omnipore Membrane filter (0.45 µm pore size, Hydrophilic PTFE membrane, 47 mm diameter), TPP™ 96-well plates, and 90 mm-diameter Petri dishes. The bacteria strains chosen as a model for Gram-positive bacteria were *Staphylococcus aureus* (ATCC 6538) and those chosen as a model for Gram-negative bacteria were *Escherichia coli* (ATCC 8739) and *Pseudomonas aeruginosa* (ATCC 9027). The strains we have chosen are considered standard microorganisms for the purpose of evaluating the antimicrobial characteristics of the newly synthesized products [44,45]. The chosen strains (*E. coli*, *P. aeruginosa* and *S. aureus*) are reference microorganisms used in the development of disinfection strategies [44,45]. The reference document for pharmaceutical products testing, *European Pharmacopoeia* (*European Pharmacopoeia* 11th edition, chapter 5.1.11), recommends these strains as reference strains for determination of bactericidal activity of antiseptic medicinal products and for bacteriostasis testing during the bacteriostasis evaluation for suitability of the method. The strains are commonly used as routine quality control strains for the verification of nutritive and selective properties of culture media used in microbiological testing for pharmaceutical products and medical devices [44].

### 2.2. Methods

#### 2.2.1. The Oxidation of Carbon Nanofibers (CNF-ox)

A mild oxidation procedure was used to improve the compatibility of the carbon nanofibers with the aqueous medium by introducing polar groups. Briefly, 2 g of carbon nanofibers (CNFs) were dispersed in 60 mL of H_2_SO_4_ (95–98%), followed by the slow dropwise addition of 20 mL of HNO_3_ (68%), under continuous stirring. The mixture was subsequently maintained under magnetic agitation for another 2 h. Then, 200 mL of H_2_O were added, and the reaction mixture was allowed to cool down before adding another 400 mL of H_2_O. In the end, the mixture was neutralized with NaOH (until reaching pH = 7) and the resulting CNF-ox were recovered by filtration followed by a thorough wash with water.

#### 2.2.2. The Synthesis of the Antimicrobial Nanoparticles and CNF–ox Decorated with Nanoparticles

##### Silver Nanoparticles (Ag-NPs)

At the beginning of this synthesis, 10 g of PVA were dissolved in 190 mL distilled water, using an Ultra-turrax^®^ disperser, at 15,000 rpm, at 70 °C, for 1 h. After cooling, 0.068 g AgNO_3_ was dissolved in the aqueous PVA solution. The pH was adjusted to 10, and while maintaining the 15,000-rpm stirring, 0.117 g of trisodium citrate and 0.018 g of NaBH_4_ were successively added. The solution was maintained under stirring for another 15 min at room temperature, and subsequently it was heated at 90 °C and maintained at this temperature under continuous stirring for another 30 min. The resulting colloidal solution containing silver nanoparticles (Ag-NPs) was stored in a dark sterile flask at 25 °C.

##### Copper Nanoparticles (Cu-Cu_2_O-NPs)

This synthesis also started by dissolving 10 g of PVA in 190 mL distilled water, using an Ultra-turrax^®^ disperser, at 15,000 rpm, at 70 °C, for 1 h. After cooling, 0.08 g Cu(CH_3_COO)_2_∙H_2_O was dissolved in this solution. The pH was adjusted to 10 by adding NH_3_, and 0.45 g of ascorbic acid was further added. The solution was maintained under stirring for another 15 min at room temperature, and subsequently it was heated at 90 °C and maintained at this temperature under continuous stirring for another 30 min. The resulting colloidal solution containing copper nanoparticles (Cu-Cu_2_O-NPs) [46] was stored in a dark sterile flask at 25 °C.

##### Zinc Oxide Nanoparticles (ZnO-NPs)

For this synthesis, 0.073 g Zn(CH_3_COO)_2_ was dissolved in 190 mL H_2_O, followed by the addition of 10 g of polyvinyl alcohol (PVA) and the complete solubilization of both components, using an Ultra-turrax^®^ disperser, at 15,000 rpm, at 70 °C, for 1 h. After cooling the solution, the pH was adjusted to 11 by adding NH_3_. The solution was maintained under stirring for another 15 min at room temperature, and subsequently it was heated at 90 °C and maintained at this temperature under continuous stirring for another 45 min. The resulting colloidal solution containing zinc oxide nanoparticles (ZnO-NPs) was stored in a dark sterile flask at 25 °C.

##### CNF-ox–Decorated with Silver Nanoparticles (CNF-ox-Ag)

0.1 g CNF-ox obtained as described above (see Section 2.2.1), were dispersed in a solution obtained by dissolving 0.068 g AgNO_3_ and 10 g of PVA in 190 mL distilled water (using an Ultra-turrax^®^ disperser, at 15,000 rpm, at 70 °C, for 1 h). After cooling the solution, the pH was adjusted to 10 by adding NH_3_. While maintaining the 15,000-rpm stirring, 0.117 g of trisodium citrate (dissolved in 3 mL of water) and 0.018 g of NaBH_4_ (dissolved in 3 mL of water) were successively added. The solution was maintained under stirring for another 15 min at room temperature, and subsequently it was heated at 90 °C and maintained at this temperature under continuous stirring for another 30 min. The resulting colloidal solution containing carbon nanofibers–decorated with silver nanoparticles (CNF-ox-Ag-NPs) was stored in a dark sterile flask at 25 °C.

##### CNF-ox–Decorated with Copper Nanoparticles (CNF-ox-Cu/Cu_2_O)

0.1 g from the CNF-ox obtained as described above (see Section 2.2.1), was dispersed in a solution obtained by dissolving 0.08 g Cu(CH_3_COO)_2_∙H_2_O and 10 g of PVA in 190 mL distilled water (using an Ultra-turrax^®^ disperser, at 15,000 rpm, at 70 °C, for 1 h). After cooling the solution, the pH was adjusted to 10 by adding NH_3,_ and 0.45 g of ascorbic acid was further added. The solution was maintained under stirring for another 15 min at room temperature, and subsequently it was heated at 90 °C and maintained at this temperature under continuous stirring for another 30 min. The resulting colloidal solution containing carbon nanofiber–decorated with copper nanoparticles (CNF-ox-Cu-Cu_2_O-NPs) was stored in a dark sterile flask at 25 °C.

##### CNF-ox–Decorated with Zinc Oxide Nanoparticles (CNF-ox-ZnO)

An amount of 0.1 g from the CNF-ox obtained as described above (see Section 2.2.1), was dispersed in a solution obtained by dissolving 0.073 g Zn(CH_3_COO)_2_∙2H_2_O and 10 g of PVA in 190 mL distilled water (using an Ultra-turrax^®^ disperser, at 15,000 rpm, at 70 °C, for 1 h). After cooling the solution, the pH was adjusted to 11 by adding NH_3_. The solution was maintained under stirring for another 15 min at room temperature, and subsequently it was heated at 90 °C and maintained at this temperature under continuous stirring for another 45 min. The resulting colloidal solution containing carbon nanofiber–decorated with zinc oxide nanoparticles (CNF-ox-ZnO-NPs) was stored in a dark sterile flask at 25 °C.

#### 2.2.3. The Preparation of the Decontaminating Formulations

The decontaminating formulations were synthesized by incorporating sodium alginate (2%, wt.%) (ALG) in the previously prepared PVA solution containing the Ag–NPs or CNF-ox–Ag-NPs colloidal solutions, with the aid of a high-shear disperser (IKA T18 digital ULTRA-TURRAX^®^, IKA-Werke GmbH & Co. KG, Staufen, Germany), until it was completely dissolved. Table 1 summarizes the sample codes of the decontaminating formulations prepared. The CNF-ox–Ag-NPs concentration was varied to observe its influence on the rheological properties of the decontaminating formulations, and further, to establish how this concentration influences the mechanical resistance of the resulting peelable films.

#### 2.2.4. The Preparation of the Nanocomposite Peelable Hydrogels

The decontaminating formulations were transferred in a Petri dish to ensure complete wetting of the surface, and a zinc acetate solution (10 wt.%) was sprayed over it to achieve the ionic crosslinking of the incorporated alginate chains (approx. 0.3 mL zinc acetate solution per 1 cm^2^ of decontaminating solution layer). Then the hydrogel films were allowed to complete the cross-linking process, and after approximately 10–15 min they could be peeled off from the surface.

#### 2.2.5. Evaluation of the Antimicrobial Activity of CNF-ox–Decorated with Nanoparticles and the Decontamination Efficacy of the Resulting Film-Forming Solutions

The antimicrobial activity of the colloidal solutions containing antimicrobial nanoparticles, and also the evaluation of the film-forming decontaminating formulations encompassing nanosized active ingredients, were performed against *Staphylococcus aureus* (ATCC 6538), *Escherichia coli* (ATCC 8739) and *Pseudomonas aeruginosa* (ATCC 9027), following the methods further described.

##### Preparing Microorganism Suspensions

Each type of microorganism was cultivated overnight in Mueller–Hinton broth at 35–37 °C with stirring (300 rpm) for 18–22 h. Then the bacterial strains were harvested. The cells were washed two times with phosphate buffer saline solution (PBS). Each time, the cells were separated using a centrifuge and resuspended in PBS. The suspensions were adjusted to approximately 10^6^–10^7^ CFU/mL, in PBS.

##### Time-Kill Test Method

The contact time approach includes exposing bacteria to the active substance being evaluated for a predetermined amount of time and at a specific antimicrobial agent concentration. Examination of the bacterial population reduction following the application of a microbicide treatment is conducted in correlation with the contact time and the concentration of the active ingredient. Portions of bacterial strain suspensions (10^7^ CFU/mL), prepared as earlier described, were treated with the colloidal solutions and with the film-forming decontaminating formulations (both at undiluted concentration), and kept at 37 °C, in direct contact, for 1 h and 24 h, respectively. At each established time, portions of bacterial cultures were serial diluted in PBS and then plated on a Muller–Hinton agar medium (MHa). After incubation at 37 °C for 24 h the bacterial survival was evaluated.

##### Minimal Inhibitory Concentration and Minimal Bactericidal Concentration

The minimum inhibitory concentration method is a quantitative method based on the technique of serial microdilutions in liquid media. The minimum inhibitory concentration (MIC) is defined as the lowest concentration of the antimicrobial substance that inhibits the visible growth of the tested microorganism (in vitro) [47]. Two-fold serial dilution in MHb of each solution was performed in duplicate. Negative and positive controls were associated. The inhibitory effect of the substances was evaluated, starting from 50% concentration (the samples of substances were diluted 1:1 with MHb). Ten µL of the microorganism suspensions (~10^4^ CFU) were added in each well corresponding to the testing samples and controls. At the end of the incubation period, 10 µL of resazurin 0.1% was added to each well. After 2 h of incubation with resazurin, the plates had been readied.

The minimum bactericidal concentration (MBC) is defined as the lowest concentration of antibacterial agent that kills 99.9% of the final inoculum after 24 h of incubation under standardized conditions [47]. MBC was determined after broth microdilution by sub-culturing the content of each well which did not show any visible signs of growth on the surface of non-selective agar plates (MHa) to determine the number of surviving cells (CFU/mL) after 24 h of incubation at 37 °C [47].

##### Cellular Viability in the Presence of the Synthesized Nanoparticles

To examine the direct impact of nanoparticles on *E. coli* (planktonic cells), portions of newly prepared bacterial suspensions in PBS, adjusted to 10^7^ CFU/mL, were placed in contact with nanoparticle solutions, achieving a final concentration of ½ (half) of the MIC value. The manufacturer’s instructions for the “LIVE/DEAD BacLight Bacterial Viability Kit” were followed when fluorescent staining was applied to the suspensions, after the pre-set contact time elapsed (35–37 °C, for 18–22 h, with constant stirring). Cells with an intact membrane (living cells) were colored green, while cells with a compromised membrane (dead or severely damaged cells) were colored red. An Olympus BX51 microscope equipped with Andor DSD2 confocal unit (fluorescence unit) was used to visualize the preparations and capture the images.

##### Efficacy of Peelable Coatings in Removing the Biological Agents from Contaminated Surfaces

The efficacy of biological contaminants removal from the tested surfaces was determined by counting the number of cells (CFUs) on the surface after hydrogel detachment. Sterile surfaces (Omnipore membrane filter placed on sterile Petri dishes) had been contaminated with known concentrations of microorganisms. Three types of microorganisms were applied separately: *Staphylococcus aureus* (ATCC 6538), *Escherichia coli* (ATCC 8739), and *Pseudomonas aeruginosa* (ATCC 9027). The contaminated filters were dried, and then the polymer solutions were applied (3 mL per filter) on the contaminated surfaces. Next, a zinc acetate solution (10 wt.%) was sprayed over the polymeric solution (approximately 5–6 mg zinc acetate/filter). The plates were kept under airflow at room temperature. After 30 min, the polymeric films formed were exfoliated. After this, the filter membrane was placed on the surface of Muller–Hinton agar to count the colony-forming units that remained on the filter and incubated at 37 °C for 24 h. The contaminated membrane filters that were only washed with PBS and were not treated with any decontaminating agent were marked as CP (positive control) in this test.

### 2.3. Characterization

Raman spectra of the pristine CNFs and oxidized CNFs were recorded on a DXR Raman Microscope (Thermo Scientific, Waltham, MA, USA) by a 473 nm laser line. The 10× objective was used to focus the Raman microscope. A high-resolution transmission electron microscope HR-TEM coupled with EDX, model TECNAI F30 G2STWIN (Fei Company, Hillsboro, OR, USA), was used at 300 kV acceleration voltage and with a resolution of 1 Å for analyzing the nanosized structures synthesized in this study (Ag, Cu-Cu_2_O, ZnO and CNF-ox-Ag, CNF-ox–Cu-Cu_2_O, CNF-ox-ZnO; all nanoparticles were dispersed in PVA solution). For the resulting decontaminating formulations, dynamic viscosity was evaluated at 25 °C, with an IKA ROTAVISC me-vi instrument, equipped with a coaxial cylinder system (O-DINS-1), and controlled via ‘Labworldsoft^®^ 6 Visc’ dedicated software. Viscosity was measured for polyvinyl alcohol-sodium alginate aqueous solutions containing different concentrations of nanofiller. The results obtained were plotted on the same multigraph, to comparatively represent dynamic viscosity as a function of shear rate. The mechanical resistance of the nanocomposite peelable films was investigated on a Discovery 850 DMA from TA Instruments (New Castle, DE, USA), equipped with TRIOS Software 5.0, by utilizing a tensile test clamp for uniaxial deformation and ‘shear-sandwich’ setup, respectively. Uniaxial tensile tests on thin films, performed on this DMA Instrument, require maintaining the axial force above the oscillation force to hold the sample in tension throughout the test. Tensile tests were carried out on rectangular specimens (50 mm × 8 mm × approximately 0.5 mm), in strain ramp mode-rate control, at 5 mm/min ramp rate. Tensile tests were performed on five specimens from each material, and the mean values were reported. Frequency sweeps for the investigation of the viscoelastic properties on the linear viscoelasticity region were performed using the ‘shear sandwich setup’ for samples measuring 10 mm × 10 mm × approximately 0.5 mm, in oscillation mode–frequency sweep. For the purpose of measuring the shear modulus, G, two identical sections of the same material were sheared between a fixed and moving plate. There was a 1% compressive pre-strain applied to nanocomposite films. The increase in frequency from 1 to 10 Hz was logarithmically scaled. Duplicate measurements were performed, and mean values were reported.

The hydrodynamic diameter and surface charge of the colloidal dispersions were characterized using a DelsaTMNano C instrument, Beckman Coulter, Istanbul, Turkey, by dynamic light scattering (DLS) and electrophoretic light scattering (ELS). Particles were illuminated by a dual 30 mW laser diode, producing time-dependent fluctuations in the intensity of the laser light. The scattered light was collected at 165° for size measurements and 15° for zeta potential measurements (diluted concentration samples), and then measured by a highly sensitive detector. Both measurement types (DLS and ELS) were performed at room temperature, each sample measurement was performed in triplicate, and for data analysis the DelsaTMNano 3.73 software was used.

## 3. Results and Discussions

To ensure good dispersion of the CNFs in aqueous media, a weak oxidation method was employed. The first stage of this study was the comparative RAMAN characterization of pristine carbon fibers and CNF-ox (Figure 1).

Figure 1 shows the RAMAN spectra for both pristine and hydrophilic CNFs. This spectroscopy analysis revealed that the I_D_ (1356 cm^−1^) to I_G_ (1575 cm^−1^) intensity ratio increased from 0.46 to 0.48 confirming the successful oxidation of the CNFs surface, without altering the structure of the carbon nanofibers [48,49].

Figure 1 presents a graphic illustration of the steps involved in the production of CNF-ox-nanoparticles.

The next step was the synthesis of the nanoparticles and CNF-ox decorated with nanoparticles. TEM-EDX analysis revealed the morphology of the synthesized nanoparticles and their elemental composition. The results are illustrated in the Appendix A and for exemplification Figure 2 below. Figure 2A,B shows that the morphology of CNF-ox did not suffer significant modification, the oxidation process leading only to minimal modification of the morphology, which is desired to ensure higher mechanical properties enhancement [50]. Appendix A displays spherical-shaped silver nanoparticles (with diameters below 50 nm), while Appendix A displays the CNF-ox-decorated with Ag-NPs. CNFs and Ag-NPs have morphological structures characterized by a tubular and apparently spherical shape. The EDX spectra from Appendix A confirmed the elemental composition of the CNF-ox-Ag-NPs. Furthermore, from both Figure 2A,B and Appendix A, it can be observed that the silver nanoparticles generated in the presence of CNF-ox are predominantly present attached to the surface of CNF-ox, mostly without aggregation, which can be explained by the presence of the polar functional groups (-carboxyl, hydroxyl and/or oxirane) [51]. The silver nanoparticles attached to the CNF-ox are predominantly round-shaped and their dimensions vary between a few nm up to 50 nm. The smaller Ag-NPs appear homogeneously dispersed, while the larger ones appear predominantly attached to the carbon nanofiber surface. Thus, the results obtained offer evidence of the successful decoration of CNF-ox with Ag-NPs and the role of the CNF-ox in particle growth and their attachment.

Appendix A illustrates the Cu-Cu_2_O-NPs, which seem to possess a homogenous size distribution (dimensions below 50 nm) but, unfortunately, they exhibited a slight predisposition to form agglomerates. Appendix A shows the CNF-decorated with Cu-Cu_2_O-NPs. Even if the Cu-Cu_2_O-NPs seem attached to carbon nanofibers, the same tendency to form aggregates was also observed in this case. The EDX spectra presented in Appendix A confirmed the elemental composition of the CNF-ox-Cu-Cu_2_O-NPs. Appendix A shows irregularly-shaped ZnO-NPs. In contrast to the Ag-NPs or Cu-Cu_2_O-NPs, ZnO particles seem to have larger dimensions (in the range 250–500 nm) and they also appear to exhibit a slight tendency to form agglomerates. Appendix A revealed that ZnO-NPs have densely and abundantly decorated CNF-ox. Appendix A shows the elemental composition of the CNF-ox-ZnO-NPs.

To assess the colloidal stability of the obtained nanoparticle dispersions, DLS analysis was performed and the zeta (ζ) potential was measured. The obtained values are presented in Appendix A and Appendix A. The obtained values confirm the presence of aggregates due to the high viscosity of PVA solutions; however, the CNF-ox act as dispersing agents, which is sustained by the smaller size of the aggregates. The zeta (ζ) potential values confirmed the stability of the obtained dispersions in all cases [52,53].

### 3.1. Antimicrobial Properties of the Synthesized Nanoparticles

#### 3.1.1. Time-Kill Test for the Synthesized Nanoparticles

The results obtained through the time-kill test for *E. coli*, *S. aureus*, and *P. aeruginosa* are illustrated in Figure 3A–C.

The time-kill test method demonstrated that the investigated compounds exhibit strong antibacterial activity against all three types of microorganisms tested, after both times of evaluations (after 1 h and after 24 h of contact) (Appendix A and Figure 3A–C). After 1 h of contact with *E. coli*, *S. aureus*, and *P. aeruginosa*, the CNF-ox-Cu/Cu_2_O nanoparticles were found to have a slightly reduced microbicidal activity than the other colloidal solutions, but still at a level high enough to allow being employed as an antibacterial agent Figure 3A–C. These results suggest that after only one hour of contact, the synthesized nanoparticles possess the ability to reduce the bacterial population (percentage of removal higher than 99.99%—Appendix A).

#### 3.1.2. Minimal Inhibitory Concentration (MIC) and Minimal Bactericidal Concentration (MBC) for the Synthesized Nanoparticles

The MIC and MBC are two important measurements for evaluating the performances of antimicrobial substances. The MIC refers to the minimum amount of such substances needed to inhibit bacterial growth following an overnight incubation, while the MBC signifies the lowest concentration required to kill a specific bacterial strain [54,55].

All the synthesized solutions demonstrated that they possess antibacterial properties, ranging from moderate to strong activity (Table 2). No MIC or MBC values were reported for the PVA solution containing only the carbon nanofibers (CNF-ox), which did not exhibit bactericidal or bacteriostatic action. A Minimum Bactericidal Concentration (MBC) value was recorded only for the colloidal solutions containing silver nanoparticles, but for the other solutions no MBC values were recorded (Table 3).

Based on the microbiology assay results, it appears that our findings align with the current literature [56,57]. From our results, it can be noticed that the AgNPs are effective against both Gram-negative and Gram-positive bacterial strains, as it is shown in the literature. Our results confirm other examples from the literature, which highlight that AgNPs exhibited strong antimicrobial effects against both Gram-positive and negative bacteria, the antibacterial activity being influenced by the physicochemical properties of AgNPs, such as size and surface, which allow them to interact or even pass through cell walls or membranes and directly affect intracellular components [58,59]. Compared to AgNPs, it can be noticed that the nano-oxides (Cu/Cu_2_O, ZnO) synthesized and tested exhibited a lower antimicrobial activity. Based on several studies, the antibacterial activity of different nano-oxides against several microorganisms could not be efficient for hindering microbial growth when not properly dispersed, which will likely be the case in cement mortars and concretes. Thus, new methods for improving the dispersion of nanomaterials are sought [60].

Relatively recent studies [61] suggest that their antimicrobial activity is dependent on the size of the silver nanoparticles, their shape, the synthetic route for their production and their MIC values, which are presented in Table 4. Another study [62], demonstrated that the MIC and MBC of 10 nm silver nanoparticles are around 1.35 mg/mL against *S. aureus.* This variation is due to the methodology used to prepare silver nanoparticles and the size of the silver nanoparticles used. The ultrafine particle size leads to an action at lower concentrations [62]. In the case of the MBC value for *E. coli*, in the literature values of between 8 and 130 mg/L are reported [63,64]. This wide range of values can be attributed to the above-mentioned elements (size, shape, and synthesis method).

The high MIC values presented by Acharya et al. [61] are due to a contact time with the stationary phase of only 4 h, compared to the standard of 18–20 h [65], but they also depend on the growth medium [66]. Consequently, the contact time influences the inhibition efficiency, but a more interesting aspect is the influence of the particle shape on the MIC value. Thus, the MIC value increases in the order of spheres (25 nm) < rods (110 nm), hexagonal (120 nm), triangle (200 nm). It can be concluded that a higher specific surface leads to a lower MIC value, assuring a higher antimicrobial effect. Even particles with the same size can present different activities, which can be attributed to a differentiation in their surface characteristics due to their synthesis route. This difference in surface characteristics affects the antibacterial mechanisms in terms of cell wall adhesion and the capacity of the silver nanoparticles to release silver ions from their surface. Relatively recent studies [67] suggest that for an important antimicrobial effect a constant rate of Ag^+^ ions migrating to the cells must be assured. Therefore, there must be a balance between silver nanoparticles’ adhesion to a support which prevents their aggregation, and the capacity for silver ions leaching in the medium, for a prolonged antimicrobial effect.

Recent studies [68,69,70] have not yet clarified in detail how the antimicrobial effect takes place, particularly the mechanism of action of the ions/nanoparticles. In the case of AgNPs, the antimicrobial effect is mostly attributed to the silver ions release from the surface of the nanoparticles, which is also the case for the Cu-NPs. The copper ions, in addition to the antimicrobial effect, after bacterial cell penetration induce the generation of reactive oxygen species. Thus, for CuNPs the process is multifaceted, with the main mechanism of bactericidal activity being the generation of reactive oxygen species, which irreversibly damages membranes [71,72]. The ZnO nanoparticles also facilitate the generation of reactive oxygen species, including hydrogen peroxide (H_2_O_2_), ∙OH (hydroxyl radicals), and O_2_^2−^ (peroxide) in the presence of light due to their semiconductor characteristics [73,74]. The antimicrobial action mechanism explanations are still under scrutiny and their findings remain an important topic in the literature, but we have selected the aspects that have received significant attention.

The same characteristics of size, shape and synthesis method also affect ZnO and Cu nanoparticles’ antimicrobial activity, as presented in Table 5 [75,76].

For copper nanoparticles the MIC and MBC values against Gram-positive and Gram-negative microorganisms are, respectively, in the range of 140–280 and 160–300 mg/L [77].

NPs act against bacterial cells by multiple pathways at the same time, which can impede the bacterial cells from acquiring resistance. Antibacterial substances in combination with NPs can become more effective against bacterial infection. Metal NPs can potently modify the metabolic activity of bacteria as evidenced by Chatzimitakos and Stalikas [78] and represent a plausible treatment of bacterial diseases. In general, small sized silver (<15 nm), gold (<10 nm), zinc-oxide (<15 nm), and titanium-oxide (<20 nm) NPs have high antimicrobial activities [79]. Since these particles act only when in contact with bacterial cell walls, various means of promoting NP-bacterial contact, such as electrostatic attraction [80], van der Waals forces [81], and receptor–ligand [82] and hydrophobic interactions [83], have been reported in the literature. NPs can cross membranes, interfere with metabolic pathways, and induce changes in the membrane shape and function. Once inside cells, NPs interact with the microbial cellular machinery to inhibit enzymes, deactivate proteins, induce oxidative stress and electrolyte imbalance, and modify gene expression levels [84].

Recent scientific research has shown that silver nanoparticles have exceptional antimicrobial properties, making them highly effective at eliminating harmful bacteria even at very low concentrations [85] of just a few milligrams per liter. Additionally, rigorous testing has demonstrated that these nanoparticles are entirely safe for eukaryotic cells, including human erythrocytes [86], so they can be utilized to combat harmful bacteria without posing any adverse health risks. Ag containing NPs can enter biofilms and prevent biofilm development by suppressing gene expression [87]. The inhibition of bacterial strain growth could be explained by specific interactions of nanoparticles with the cell envelope of micro-organisms [88]; for example, nanoparticles that can enter the cell membrane interact with bacterial enzymes, damaging the cell [89]. Some nanoparticles interact electrostatically with the bacterial membrane, and reactive oxygen species are generated, leading to disruption of the membrane and DNA damage [90].

The bactericidal characteristics of AgNPs are dramatically influenced by particle shape, size, concentration, and colloidal state [91,92]. Smaller AgNPs sizes appear to increase biocompatibility and stability [93,94]. For example, 5–10 nm sized AgNPs exhibited bacteriostatic and bactericidal activity against *S. aureus*, methicillin-susceptible *S. aureus* (MSSA), and methicillin-resistant *S. aureus* (MRSA) [95]. In addition to their antimicrobial properties, silver nanoparticles have been found to be particularly beneficial in preventing the development of antibiotic-resistant bacteria [96], which has become a major global health concern in recent years. As a result, silver nanoparticles have been identified as a promising alternative to conventional antibiotics in the fight against bacterial infections [96]. In our case, according to the outcomes from the microbiology tests (Table 2 and Table 3, and Appendix A), the aqueous PVA solutions containing silver (Ag-NPs and CNF-ox-Ag-NPs) possess a strong bactericidal action against Gram-negative and Gram-positive bacteria. Moreover, silver–decorated with CNFs-ox led to lower MIC values than the analogous Ag-NPs, meaning that the combination of CNF-ox and Ag-NPs ensures a higher inhibitory effect due to the lower particle size in the case of CNF-ox-Ag-NPs [97,98].

Due to its claimed low toxicity to the human body and antibacterial properties, copper has recently attracted the attention of scientists [99]. For instance, the copper nanoparticles are efficient against both Gram-negative and Gram-positive bacteria but also against fungal species [100] or multi-resistant microorganisms [99,101]. It appears that the efficacy of copper is influenced by its concentration, potentially leading to either a bacteriostatic or bactericidal effect [101]. Copper antimicrobial activity is also influenced by the dimensions of the copper nanoparticles (higher specific surface areas), which should allow a higher surface of contact between the nanostructures and the bacterial cell [99] to obtain superior decontaminating performances. Our solutions containing copper nanoparticles exhibited significant antibacterial activity against Gram-negative bacteria, but only moderate antibacterial activity against Gram-positive bacteria (Table 2, Appendix A).

The mechanisms responsible for the antimicrobial activity of ZnO-NPs have yet to be clarified. Proposed mechanisms include (1) the destruction of cell integrity caused by direct contact between ZnO-NPs and cell walls [19], (2) ROS formation [20], and (3) the release of antimicrobial ions, mainly Zn^2+^ ions [19]. However, because the chemical nature of dissolved zinc depends on media constituents, it is likely that the mechanism of ZnO-NP toxicity is media dependent [19]. It has been reported that ZnO may have a lower level of toxicity when compared to silver nanoparticles [102]. However, ZnO-NPs are also able to enter the bacterial cell and release metal ions, which can cause damage to bacterial DNA [102]. Additionally, ZnO can generate reactive oxygen species (ROS) [103,104,105]. Surface imperfections on ZnO abrasive texture [73] can also improve its antibacterial effects. In our case, in contrast with silver and copper—based nanostructures developed in this study, the activity of the colloidal solutions containing zinc oxide proved to be to some extent lower (Table 2, Appendix A).

Correlating the results obtained, we can affirm that the newly manufactured nanoparticles have strong antibacterial activity, both in concentrated form and in diluted solutions.

#### 3.1.3. Effect of Nanoparticles and CNF-Based Nanostructures on Cell Viability

The effect of antimicrobial agents on a bacterial population can be evaluated qualitatively by examining microscopic preparations stained with specific dyes and visualized in fluorescence mode. The coloration differentiates between living and dead cells using the live-dead staining technique. This technique is used to assess the viability of bacterial populations based on the integrity of their cell membranes. By using this method, valuable information about a compound’s toxicity to the bacterial population can be obtained [106]. In the current work, we used this approach to evaluate *E. coli* interaction with all the synthesized colloidal solutions.

The images (Figure 4A–G) display living (green color) and dead (red color) bacterial cells for the *E. coli* bacteria put in contact with the antimicrobial solutions (containing the active nanoparticles or CNF-decorated with nanostructures) at concentrations of ½ MIC. These images demonstrate that the bacterial cells were significantly impacted by the nanoparticles, which led to a significant reduction. It was found that solutions containing copper and silver nanoparticles had the most significant impact on cell viability, whereas solutions containing zinc had a minimal effect on the bacterial population.

The results of the microscopic evaluations are in accordance with the data obtained in the MIC determination method. These preliminary findings are encouraging, but additional studies and experiments will probably help to fully understand how these nanoparticles affect bacterial cells. Since from all the colloidal antimicrobial solutions tested, the information obtained indicated the best outcomes for silver, only combinations of CNF-ox–Ag-NPs were further investigated in the next experiments. To obtain the decontaminating formulations, sodium alginate was further added to the colloidal solutions containing the silver-based nanostructures. More details for the preparation of the decontaminating formulations were described in Section 2.2.3 and their rheological properties are depicted below.

### 3.2. Rheological Properties

#### Dynamic Viscosity of the Decontaminating Formulations

Investigating the rheological behavior of the decontaminating formulations is important when evaluating the options available for applying them on the contaminated surfaces (with the aid of a spraying device, by brush, by roller, etc.). Figure 5 illustrates the viscosity as a function of shear rate, measured for polyvinyl alcohol-sodium alginate aqueous solutions containing different concentrations of nanofiller.

All the samples exhibited a non-Newtonian pseudoplastic (shear thinning) behavior. As a result of the shear stress applied, the molecules tend to rearrange, to reduce the overall stress. The ‘blank’ sample, containing only polyvinyl alcohol and sodium alginate registered the lowest viscosity values. The intermolecular hydrogen bonds [107,108] influence the rheological behavior of the decontaminating formulations. Through the oxidative pre-treatment, polar functional groups (hydroxyl, carboxyl, and/or oxirane) [51] were introduced on the surface of CNFs-ox, allowing more hydrogen bonds to form between the CNFs-ox and the polymer chains, while improving dispersion in the polymeric aqueous solution and reducing the probability of nanofiber agglomerations. The presence of the nanofibers in the polyvinyl alcohol—sodium alginate aqueous solutions leads to higher viscosities than the spherical-shaped silver nanoparticles because the particle motion is geometrically hindered [109] in the case of CNFs-ox. The viscosity increases with the content of CNF-ox, which leads to an increase in intermolecular interactions (hydrogen bonds) between the components (CNF-ox, alginate, and PVA).

### 3.3. Mechanical Properties

The decontaminating formulations were crosslinked with a zinc acetate solution to investigate the effect of CNF concentration on the mechanical properties of the resulting nanocomposite peelable films. The mechanical properties of these peelable coatings are critical for decontamination applications because they should be flexible enough to be peeled off as a continuous sheet after serving their purpose. Our peelable nanocomposite films underwent tensile testing and frequency sweep measurements (on the linear viscoelasticity region, using a ‘shear-sandwich’ configuration), to assess their mechanical resistance and viscoelastic behavior, respectively. The results obtained are listed below.

### 3.4. Evaluation of the Viscoelastic Properties of the Peelable Films Reinforced by Carbon Nanofiber—Decorated with Antimicrobial Nanoparticles

In general, frequency sweeps are used to characterize the non-destructive deformation behavior of a sample over time. Low frequencies are used to simulate slow motion, whereas high frequencies are used to represent fast motion in short timeframes. The viscoelastic properties of the specimens analyzed can be compared from plots of the storage or loss moduli illustrated in Figure 5.

The results summarized in Figure 6 confirm a well-structured (gelled) system. The components are strongly associated; therefore, the storage modulus (G′) is greater than the loss modulus (G″), and both are to some extent almost frequency independent. The storage modulus is a measure of the elastic component of the material being studied, and the loss modulus is a measure of its viscous component. For samples of PVA-ALG-c1-CNF-Ag, PVA-ALG-c2-CNF-ox-Ag, and PVA-ALG-c3-CNF-Ag it was observed that the storage modulus increases as the nanofiller content increases, probably due to the elastic component dominance indicating the absence of phase separation of these specimens. In contrast to samples PVA-ALG-c2-CNF-Ag and PVA-ALG-c3-CNF-Ag, lower G′ values registered for PVA-ALG-c4-CNF-Ag indicate that the higher concentration of CNFs may have caused some agglomeration of the nanofiller. Thus, probably due to the aggregated nanofibers, the polymeric network structure will undergo destruction and reconstruction, resulting in fluctuations of viscoelastic characteristics [110] and a slight suppression of elastic behavior [111,112]. G″ describes the deformation energy that is lost by internal friction during shearing [113] and which exhibits similar patterns for all samples. The slight increase in loss modulus values implies enhanced damping due to the increase in interfacial surface area resulting from high aspect ratios of CNFs-ox [112]. Increased nanofiller concentration can be beneficial for improved energy dissipation capabilities, but an agglomeration of nanoparticles could influence the damping properties of the composite [112].

### 3.5. Tensile Test Performed on the Peelable Films Reinforced by Carbon Nanofiber–Decorated with Antimicrobial Nanoparticles

Tensile strength is an important feature for peelable coatings to allow peeling from the substrate in a single continuous sheet. Figure 7 illustrates comparative plots highlighting the influence of the nanofiller concentration on the mechanical performances of the nanocomposite peelable films designed for decontamination applications.

All the films obtained in this study possessed the appropriate mechanical resistance for being peeled as a continuous sheet [13,114]. However, to optimize this capacity the comparative plots in Figure 7 help establish the optimal concentration of CNF for this type of application. As can be observed from Figure 7A, the presence of the nanofillers had a positive influence on the tensile strength of the peelable films PVA-ALG-Ag, PVA-ALG-c1-CNF-Ag, and PVA-ALG-c2-CNF-Ag. Nevertheless, at CNFs-ox concentrations higher than 0.05% (samples PVA-ALG-c3-CNF-Ag and PVA-ALG-c4-CNF-Ag) the ultimate tensile stress values registered a reduction (Figure 7B), probably due to the tendency towards forming aggregates inside the composite matrix at higher nanofiller contents.

Therefore, after evaluating the viscoelastic behavior and the tensile strengths of the synthesized nanocomposite films it can be affirmed that the sample PVA-ALG-c2-CNF-Ag, containing 0.05% CNFs, possesses the optimal characteristics for this application.

### 3.6. Decontamination Method Principle

Figure 2 contains an illustrative description of the decontamination process.

The method of decontamination involves a straightforward procedure in two simple stages: coverage of the contaminated area with the decontaminating formulation (Figure 2—step marked with 2), and, after allowing approximately 15–20 min for the active nanoparticles to interact with the pathogen microorganisms (Figure 2—step marked with 3), the second phase (Figure 2—step marked with 4) will consist in applying a bivalent ion-containing solution (in this case zinc acetate, because it can enhance the antimicrobial effect) to ensure the formation of a continuous, mechanically resistant and easily peelable nanocomposite polymeric film which will entrap the contaminants, thus obtaining a decontaminated surface (Figure 2—final step, marked with 5). Table 6 summarizes the components of the decontaminating formulations.

### 3.7. Antimicrobial Properties of the Nanocomposite Peelable Films

The second antimicrobial assay involved a time-kill test and MIC/MBC evaluations for the decontaminating formulations, but it also focused on investigating the decontamination performances of the nanocomposite peelable films by evaluating their efficacy in removing biological agents from contaminated surfaces.

#### 3.7.1. Time-Kill Test for the Polymeric Decontaminating Formulations Containing Ag-NPs and CNF-ox Ag-NPs

The time-kill test (Figure 8A–C) for the decontaminating formulations was performed on the same type of bacterial strains, *E. coli*, *P. aeruginosa*, and *S. aureus*, as per the analogous colloidal solutions that were previously tested. This test was also executed for the polymeric decontaminating formulations, to establish if the active ingredients still maintain their antimicrobial activity when embedded in a film-forming polymeric blend based on PVA and ALG. After the proposed contact times (1 h and 24 h) between the bacterial strains and the decontaminating formulations, the bacterial growth was evaluated.

The time-kill assay showed that even after 1 h of contact, the decontaminating formulations containing Ag-NPs and CNF-ox-Ag-NPs (Component A) showed strong activity against the three microorganisms tested, while the blank solution PVA-ALG or the solution containing only carbon nanofibers (PVA-ALG-c2-CNF-ox) did not reveal any antibacterial activity. Zinc acetate solution (Component B, Table 5) also revealed a high antibacterial activity against all three microorganisms tested. The decontaminating formulations that had a high level of antimicrobial activity after one hour also had a significantly high level of antimicrobial activity after 24 h, in contrast to the other evaluated solutions, which had no antimicrobial activity even now (PVA-ALG and PVA-ALG-c2-CNF-ox). The strong antimicrobial activity can be attributed to the presence of silver in the decontaminating formulations containing Ag-NPs (sample code PVA-ALG-Ag-NPs) or CNF-ox-Ag-NPs (sample code PVA-ALG-c2-CNF-Ag).

#### 3.7.2. Minimal Inhibitory Concentration (MIC) and Minimal Bactericidal Concentration (MBC) for the Polymeric Decontaminating Formulations Containing Ag-NPs and CNF-ox-Decorated with Ag-NPs

The MIC and MBC values obtained for the polymeric decontaminating formulations containing Ag-NPs and CNF-ox-decorated with Ag-NPs are shown in Table 7. The decontaminating formulations containing silver nanoparticles or CNF-decorated with silver nanoparticles showed the strongest antimicrobial activity against Gram-negative bacteria (*E. coli* and *P. aeruginosa*) and a pronounced activity against Gram-positive bacteria (*S. aureus*). In contrast, MIC values for samples PVA-ALG and PVA-ALG-c2-CNF revealed the lowest antimicrobial activity against the bacterial strains used in the test. MBC values were revealed for samples PVA-ALG-Ag and PVA-ALG-c2-CNF-Ag, but not for samples PVA-ALG and PVA-ALG-c2-CNF, in the conditions of the test. The samples without MBC values had bacteriostatic activity, but no bactericidal activity.

#### 3.7.3. Efficiency of the Decontaminating Peelable Films Containing Ag-NPs and CNF-Decorated with Ag-NPs in Removing Biological Agents from Contaminated Surfaces

To evaluate the capacity for entrapping the biological contaminants, a slightly porous surface (described in Section 2) was chosen to perform decontamination tests, thus verifying if the decontaminating formulations can efficiently remove the bacterial strains. Each test was performed in five replicates/microorganisms. The targeted surfaces were initially subjected to a controlled contamination procedure (described in Section 2.2) followed by the application of the decontaminating formulations on the contaminated surfaces (Figure 9A). Subsequently, the zinc acetate solution (Component B) was sprayed over the samples already covered by Component A. After 30 min, the polymeric films formed were exfoliated (Figure 9B). The removal efficacy was estimated by counting the residual colony forming units after the decontaminated surface was incubated at 37 °C for 24 h on Muller–Hinton agar. The CFU values registered after decontamination were compared to the initial CFU values, to obtain the decontamination efficacy (Table 8).

With one exception (blank sample, PVA-ALG, for *P. aeruginosa*), the percentage of bacteria removed exceeded 93%. Moreover, the peelable films reinforced by the CNF-ox decorated with Ag-NPs led to the highest values in terms of decontamination efficacy, thus demonstrating the synergistic effect of the active ingredients.

This innovative approach ensures an efficient decontamination process by combining the entrapment abilities—useful for sequestering the bacterial strains inside the polymeric matrix with the powerful bacteriostatic and bactericidal effect—exerted by the active nanoparticles incorporated in the decontaminating formulations.

All the performed microbiology tests have shown that the presence of CNF-decorated with Ag-NPs in the decontaminating formulations brings significant advantages to the decontamination process. In comparison to Ag-NPs, a higher amount of silver nanoparticles with considerably smaller dimensions were generated in the presence of the carbon nanofibers (Appendix A). Due to their notably smaller dimensions, the silver nanoparticles synthesized in the presence of the CNFs-ox can migrate more efficiently toward the bacterial strains, as can be observed in Figure 10. Thus, in the case of PVA-ALG-c2-CNF-Ag samples, besides the fact that they ensure a higher specific surface area available for interactions with the bacterial cells, some of the silver nanoparticles can exit the ionically crosslinked polymeric network and approach the contaminant, thus ensuring a more efficient contact with the microbial strains.

Based on the results obtained, we can affirm that the polymeric nanocomposite films containing Ag-NPs, and especially the ones incorporating the CNF-ox-Ag-NPs, represent a promising tool for biological decontamination of surfaces.

## 4. Conclusions

This study presents the synthesis of decontaminating polyvinyl alcohol-alginate hydrogel films containing carbon nanofibers and antimicrobial nanoparticles. CNF modification was demonstrated by RAMAN spectroscopy. Antimicrobial particles were synthesized by chemical reduction (silver and copper) and precipitation (ZnO) in the presence of PVA as a stabilizer and PVA and CNF. The presence of CNF-ox during the nanoparticle synthesis leads to smaller particle size which can be attributed to CNF-ox acting as an anchoring point for the growth of the nanoparticles due to the polar functional groups on their surface. The nanoparticles morphology was investigated by TEM, and their elemental composition by EDX. DLS analysis confirmed the role of CNFs to improve the dispersion of the NPs, leading to a higher stability of the dispersions. The antimicrobial activity experiments revealed a higher activity for the CNF-ox decorated with nanoparticles, the highest activity being registered for silver nanoparticles. The influence of the CNF-ox content on the mechanical properties of the Zn^2+^ crosslinked hydrogels was evaluated to determine the optimum amount of reinforcing agent, a higher content leading to a decrease in the tensile resistance due to aggregation.

Finally, the antimicrobial activity and biological decontamination capacity of the zinc acetate cross-linked hydrogels were evaluated for *S. aureus*, *P. aeruginosa*, and *E. coli* contamination. The results confirm the good capacity of the hydrogels to create a sterile surface within minutes, while the biological contamination trapped in the polymeric material is inactivated after 24 h, reducing the possibility of cross-contamination of other areas.

The decontamination efficiencies against *Staphylococcus aureus*, *Escherichia coli*, and *Pseudomonas aeruginosa* varied between 97.40% and 99.95% for the silver-decorated CNF-ox based formulations. The results confirmed a higher antimicrobial activity explainable by a collaborative effect of silver and CNF-ox, coupled with the efficient entrapment of the contaminants inside the peelable films.

## Data Availability

Not applicable.

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
