# Peer review of "Peelable Alginate Films Reinforced by Carbon Nanofibers Decorated with Antimicrobial Nanoparticles for Immediate Biological Decontamination of Surfaces"

_nanomaterials, 2023, doi:10.3390/nano13202775_

Round 1
Reviewer 1 Report
The article contains the interesting results of experiments with nanocomposite films that are promising for the disinfection of open surfaces. The authors conducted comprehensive research of the synthesized films and also tested the bacterial activity of different components. A wide arsenal of modern methods was used in the work, which made it possible to study the properties of nanocomposite films, as well as their antimicrobial activity. The description of the methods is quite detailed, which makes it possible to get a complete picture of the research. Tables and illustrations quite clearly reflect the data obtained. In general, the work is novel and points to the practical prospects of the proposed approaches in solving the problems of surface decontamination.
Remarks.
1) The abstract describes in detail the methods used in this study. The same applies to the conclusion. This description should be made more concise, but the most important results should be added and their significance noted.
2) The first three paragraphs in the introduction look overly detailed (until line 87). Decontamination of exposed surfaces is an obvious problem.
3) Three bacterial strains were chosen as test objects. Two of them are Gram-negative and one is Gram-positive. Obviously, it would be nice to comment the choice of these strains. In a number of studies, bacteria of the genus Bacillus (B. subtilis, B. cereus), as well as the yeast fungus Candida albicans, are also used for these purposes. In addition, there are no any characteristics of the chosen strains (except for registration numbers). It would be important to know whether these strains were previously used for such studies, whether their resistance to biocidal formulations was studied, etc.
4) It is not entirely clear why the minimal inhibitory concentration was determined for all solutions (Table 2), while the minimal bactericidal concentration was determined only for colloidal solutions containing silver nanoparticles (Table 3). Why this concentration has not been determined, for example, for solutions containing copper nanoparticles.
5) 484-486. The authors write: Since from all the colloidal antimicrobial solutions tested, the information obtained indicated the best outcomes for silver, only combinations of CNF-ox-AgNPs were further investigated in the next experiments.
However, it follows from Figure 4 that the largest number of dead cells was obtained in the variant with copper. It needs to be explained.
6) It is not very clear why the direct effect of nanoparticles on bacterial cells was studied only on the example of Escherichia coli, and not on all three strains. This needs clarification.
Moderate editing of English language required.
Author Response
Dear reviewer,
Thank you. The answers are in attached file.

Reviewer 2 Report
The study presents an important contribution to the field of decontamination through the development of peelable coatings using alginate films reinforced with carbon nanofibers decorated with antimicrobial nanoparticles. I appreciate the effort put into the experimental work and the clarity of the results. However, there are a few questions that require further clarification and improvement. Please find a detailed review below:
1. In the introduction, you mention the need for fast and efficient decontamination methods. Considering the materials and techniques used in this study, how do the developed peelable alginate films compare to other existing decontamination methods in terms of speed and efficacy? It would be helpful to discuss any limitations or challenges that need to be addressed for practical implementation.
2. The study primarily focuses on the remediation of biological contamination. Could you discuss the potential applications of the developed peelable alginate films in other fields apart from biological decontamination? Are there any additional modifications or adaptations that could broaden the scope of their use?
3. The format of the references needs to be consistent, e.g. some journal names are in type of full name while others use abbreviations.
Author Response

(The authors gave the same response as above.)

Reviewer 3 Report
Antibacterial nanostructured hydrogels have attracted the attention of researchers and have good potential for practical application. The work sounds interesting and has good scientific quality. To improve the manuscript, the authors should add the results of studies of particle sizes in colloidal systems and the stability of suspensions, as well as discussing the following recent articles:
El Fadl FIA, Hegazy DE, Maziad NA, Ghobashy MM. Effect of nano-metal oxides (TiO2, MgO, CaO, and ZnO) on antibacterial property of (PEO/PEC-co-AAm) hydrogel synthesized by gamma irradiation. Int J Biol Macromol. 2023 Aug 9;250:126248. doi: 10.1016/j.ijbiomac.2023.126248.
Dong Q, Zu D, Kong L, Chen S, Yao J, Lin J, Lu L, Wu B, Fang B. Construction of antibacterial nano-silver embedded bioactive hydrogel to repair infectious skin defects. Biomater Res. 2022 Jul 25;26(1):36. doi: 10.1186/s40824-022-00281-7.
Pan Y, Li P, Liang F, Zhang J, Yuan J, Yin M. A Nano-Silver Loaded PVA/Keratin Hydrogel With Strong Mechanical Properties Provides Excellent Antibacterial Effect for Delayed Sternal Closure. Front Bioeng Biotechnol. 2021 Oct 8;9:733980. doi: 10.3389/fbioe.2021.733980.
Author Response

(The authors gave the same response as above.)

Reviewer 4 Report
The work reported by the authors is interesting and worth publishing.
Some revisions are necessary before publication. Data regarding the antibacterial activity are not presented in a clear way; the histograms of Figure 3 should be vertical and not horizontal. Moreover, in this format, it is difficult to follow the beahviour of the data with time; it will be better to show the activity for each sample with time separately - see for instance Figure 7, J. Phys. Chem. C, 118, 4751, 2014. The same should be done for Figure 8.
Also, in these graphs the legend of the Y axis is "log(CFU)"; in this case the values should be between for instance -1 and +8, not between 10^-1 and 10^+8.
Finally, and more important, the authors do not perform ant statistical analysis on these data; this should be added.
In figure 4, there is no dimension bar on the images.
Overall, the authors should explain better the element(s) of novelty of this work in relation to the research already published.
Minor revision of the language should be performed.
Author Response

(The authors gave the same response as above.)

Reviewer 5 Report
The manuscript "Peelable alginate films reinforced by carbon nanofibers decorated with antimicrobial nanoparticles for immediate decontamination of surfaces" presents a study on the synthesis of decontaminating polyvinyl alcohol-alginate hydrogel films containing carbon nanofibers (CNFs) and antimicrobial nanoparticles. The synthesized nanoparticles were characterized using TEM-EDX analysis, and the antimicrobial properties were evaluated through time-kill tests, MIC/MBC assays, and decontamination efficacy tests. The rheological and mechanical properties of the resulting nanocomposite peelable films were also investigated. Overall, the manuscript provides valuable insights into the synthesis and characterization of decontaminating polymeric films with antimicrobial properties. The inclusion of carbon nanofibers and various types of nanoparticles enhances the antimicrobial activity of the films, especially those containing CNF-ox decorated with nanoparticles. The results also demonstrate the successful decontamination of surfaces using these films. However, there are several areas that need improvement in the manuscript:
1. The introduction section lacks a clear and concise statement of the problem being addressed and the significance of the research. It would be helpful to provide a brief background on the importance of decontamination and the limitations of existing methods.
2. The methods section needs more details and clarity regarding the synthesis and characterization of the nanoparticles, the preparation of the polymeric films, and the evaluation of their antimicrobial and mechanical properties. The readers should have all the necessary information to replicate the experiments.
3. The results section should be organized better to clearly present the findings. It is currently difficult to follow the sequence of the experiments and their corresponding results.
4. The discussion section needs to be expanded to provide a more thorough analysis and interpretation of the results. The authors should discuss the implications of their findings, relate them to existing literature, and highlight the novelty and contributions of their research.
5. The conclusion section is too brief and does not adequately summarize the key findings and implications of the research.
6. The manuscript would benefit from a thorough proofreading to correct grammatical errors and improve the clarity of the language.
For example:
-Inconsistent in "PVA" (also written as "pva") and "AgNPs" (also written as "Ag-NPs" and "Ag NPs")
-Typographical errors in:
"bactericidal" (misspelled as "bactericial")
"decontaminanating" (should be "decontaminating")
"morphology" (misspelled as "morpholgy")
"fluorescence" (misspelled as "fluorescense")
"strenght" (should be "strength")
Overall, the manuscript has potential but requires major revisions to address the above-mentioned issues. With the necessary revisions, the study can make a valuable contribution to the field of decontamination and antimicrobial materials.
Author Response

(The authors gave the same response as above.)

Reviewer 6 Report
1.Compared with Applied Surface Science ,2018,454, 101-111;Applied Surface Science ,2018,445, 39-49,the noverty of the manuscript should be emphasised.
2.The exitence of Ag–NPs is insufficent.The authors should add more detais to the manscript.
3.The mechanism of antimicrobial activity is too simple.The authors should add more detais.
Minor editing of English language required
Author Response

(The authors gave the same response as above.)

Round 2
Reviewer 4 Report
The authors addressed the points raised by the reviewers and the manuscript can be published with no additional changes.
Minor revision of the English is advised.
Author Response
Thank you very much. The English was polished.
Reviewer 5 Report
I have reviewed the manuscript titled "Peelable Alginate Films Reinforced by Carbon Nanofibers Decorated with Antimicrobial Nanoparticles for Immediate Decontamination of Surfaces" authored by Gabriela Toader et al. The manuscript describes the synthesis and characterization of alginate-based nanocomposite peelable films with promising antimicrobial properties. I am pleased to report that all suggested changes and revisions have been adequately addressed, significantly enhancing the clarity and scientific rigor of the manuscript. I believe that the manuscript is now suitable for publication in the journal "Nanomaterials."
The manuscript is well-structured, and the research conducted is both relevant and timely, as it addresses the development of materials with immediate decontamination potential. The abstract provides a concise summary of the study, and the introduction sets a clear context for the research. The methodology is robust and thoroughly explained, and the results are presented logically with appropriate figures and tables. The discussion section effectively interprets the findings and their implications, and the conclusion summarizes the key outcomes.
Overall, the manuscript represents a well-executed and scientifically rigorous study with the potential for significant impact in the field of nanomaterials and surface decontamination. I commend the authors on their work and look forward to seeing it published.
Author Response
Thank you very much.